# Coordinative Chain Transfer Polymerization of Sustainable Terpene Monomers Using a Neodymium-Based Catalyst System

**DOI:** 10.3390/polym14142907

**Published:** 2022-07-17

**Authors:** Teresa Córdova, Francisco Javier Enríquez-Medrano, Eduardo Martínez Cartagena, Arnulfo Banda Villanueva, Luis Valencia, Edgar Nazareo Cabrera Álvarez, Ricardo López González, Ramón Díaz-de-León

**Affiliations:** 1Research Center for Applied Chemistry, Enrique Reyna Hermosillo, No.140, Col. San Joseé de los Cerritos, Saltillo 25294, Mexico; trscordova@gmail.com (T.C.); javier.enriquez@ciqa.edu.mx (F.J.E.-M.); martinezme.d17@ciqa.edu.mx (E.M.C.); abanda.d18@ciqa.edu.mx (A.B.V.); 2Biofiber Tech Sweden AB, Norrsken Hourse, Birger Jarlsgatan 57 C, SE-11356 Stockholm, Sweden; luisalex_val@hotmail.com; 3Research Center for Applied Chemistry Monterrey Unit, Av. Alianza Sur No.204, Apodaca 66629, Mexico; edgar.cabrera@ciqa.edu.mx

**Keywords:** myrcene, farnesene, biobased monomer, coordinative chain transfer polymerization

## Abstract

The present investigation involves the coordinative chain transfer polymerization (CCTP) of biobased terpenes in order to obtain sustainable polymers from myrcene (My) and farnesene (Fa), using the ternary Ziegler–Natta catalyst system comprising [NdV_3_]/[Al(*i*-Bu)_2_H]/[Me_2_SiCl_2_] and Al(*i*-Bu)_2_H, which acts as cocatalyst and chain transfer agent (CTA). The polymers were produced with a yield above 85% according to the monomeric consumption at the end of the reaction, and the kinetic examination revealed that the catalyst system proceeded with a reversible chain transfer mechanism in the presence of 15–30 *equiv.* of CTA. The resulting polyterpenes showed narrow molecular weight distributions (M_w_/M_n_ = 1.4–2.5) and a high percent of 1,4-*cis* microstructure in the presence of 1 *equiv*. of Me_2_SiCl_2_, having control of the molecular weight distribution in Ziegler–Natta catalytic systems that maintain a high generation of 1,4-*cis* microstructure.

## 1. Introduction

The environmental need to reduce greenhouse emissions and the consumption of fossil resources has promoted the development of more ecofriendly materials, including biobased elastomers [1,2,3]. In this context, the search for alternative biobased monomers that can (at least partially) replace the butadiene or isoprene monomers in the production of synthetic rubbers, which are obtained from steam cracking, is crucial. A prominent alternative is the use of biobased terpenes as monomers, which are natural hydrocarbons present in renewable resources such as vegetable resins and essential oils [4,5,6]. Most terpenes contain the isoprene group (CH_2_=C(CH_3_)-CH=CH_2_) in their chemical structure, and, therefore, they are capable of being polymerized to give rise to the formation of materials with elastomeric properties. However, the successful polymerization of terpenes is not as easy as it sounds since conventional reaction conditions (by diverse addition polymerization methods) lead to low molecular weight and broad molecular weight distributions (MWDs) [7,8,9,10]. Therefore, what polymerization pathway is the most appropriate for this family of monomers is still uncertain, but it represents a major market potential considering the possibility of developing bio-rubbers with reduced carbon footprints. 

Some of the most abundant terpenes, such as β-pinene and limonene, have been reported to be polymerizable principally by free radicals [11,12,13,14] and cationic [15,16,17,18,19] mechanisms, resulting, in most cases. in materials with very low molecular weights as a result of steric effects caused by their bulky structures [20]. Thus, the synthesis of novel terpene-based rubbers with adequate properties in order to have competitive performance (which requires high enough molecular weights and stereocontrol) is required. 

The growing interest in rare earth metal-based catalytic systems for the polymerization of 1,3-conjugated dienes to produce high-performance rubbers is due to their high catalytic activities and a high degree of stereocontrol [21,22,23,24,25]. However, the obtained polymers using these types of catalytic systems are characterized by broad MWD because of the formation of multiple catalytic species with different kinetics, and also because of the irreversibility of chain transfer reactions that occur throughout the polymerizations. Nevertheless, by altering some reaction conditions, especially the cocatalyst/catalyst and monomer/catalyst ratios, it has been proven to be feasible to change the irreversibility of the chain transfer reaction, giving way to a CCTP mechanism. This mechanism confers living features to the polymerization, as well as the possibility of synthesizing block copolymers via sequential addition; this is based on the fact that the employed cocatalyst is also able to act as a chain transfer agent (CTA). The CCTP of butadiene [26,27,28,29,30] and isoprene [31,32,33,34] has been carried out with very promising results. The potential of rare earth metal-based catalytic systems in regular (irreversible) coordination polymerizations has been successfully extended to the stereoregular polymerization of terpenes, such as myrcene (My), farnesene (Fa), and (E)-α-ocimene (Oc) [35,36,37,38]. Nevertheless, CCTP conditions for this family of renewable monomers have scarcely been reported to date.

Georges et al. previously reported the polymerization of My using the Cp*La(BH_4_)_2_(THF)_2_ (Cp* = pentamethylcyclopentadienyl) complex as a catalyst in combination with n-butylethylmagnesium and triisobutylaluminum under CCTP conditions to yield high 1,4-*trans* polymyrcene as a product. [Mg]/[Al]/[La] = 1/9/1, 1/19/1, and 1/39/1 were the evaluated ratios, which had a significant influence over the molecular weight of the polymers, but not on the dispersity values (Ð), nor the 1,4-*trans* content or *T_g_* values. They furthermore explored the statistical copolymerization of My, successfully reporting the synthesis of copolymers poly(myrcene-*co*-styrene), poly(myrcene-*co*-isoprene), and poly(myrcene-*co*-isoprene-*co*-styrene) [39]. Loughmari et al. reported the use of the catalyst system Nd(BH_4_)_3_(THF)_3_/*n*-butylethylmagnesium for My polymerization. They found CCTP conditions (good catalytic activities, relatively low Ð, and good agreement between the theoretical and experimental average number molecular weight) at [Mg]/[Nd] ratios > 1, and the best results at ratios 2 to 5 [33]. Of note, the aforementioned works are among the few reported in the literature regarding My polymerization under CCTP conditions. Moreover, to the best of our knowledge, there are no further reports related to the CCTP of Fa as a monomer or comonomer.

In this work, we systematically studied the polymerization of myrcene (My, 7-methyl-3-methylene-octa-1,6-diene) and farnesene (Fa, 3,7,11-trimethyldodeca-1,3,6,10-tetraene) (Figure 1) via coordinative chain transfer polymerization (CCTP) and compared it with isoprene (Ip). The polymerizations were carried out via a homogeneous ternary catalytic system based on neodymium versatate/diisobutylaluminum hydride/dichlorodimethylsilane (NdV_3_/Al(*i*-Bu)_2_H/Me_2_SiCl_2_) for Ip, My, and Fa polymerizations under CCTP conditions, which has been previously reported for the polymerization of butadiene [30], isoprene [32,40], butadiene/myrcene [41], and butadiene/farnesene [41] with promising results. The influence of multiple reaction variables, such as reactant molar ratios (catalyst/cocatalyst/activator) and reaction temperature, were systematically studied with respect to the polymerization behavior of the biobased diene monomers, as well as the final properties of the resultant polyterpenes in terms of molecular weight characteristics, microstructure, and thermal properties. The controlled nature of the system was evaluated through chain extension reactions.

## 2. Materials and Methods

The manipulations of all reagents were carried out under an inert atmosphere using an MBraun glovebox (Labmaster 130, MBraun, Gaithersburg, MD, USA) or using a dual vacuum–argon line and standard Schlenk techniques. The monomers isoprene (Ip, 99%, from Sigma-Aldrich, St. Louis, MO, USA), β-myrcene (My, ≥90%, from Ventós S.A., Barcelona, Spain), and trans-β-farnesene (Fa, 93–95%, from Amyris, Inc., Emeryville, CA, USA) were distilled from sodium under an argon atmosphere before use. The catalyst neodymium versatate (NdV_3_) was obtained from Solvay as a 0.54 M solution in hexanes (Brussels, Belgium); diisobutylaluminum hydride (Al(*i*-Bu)_2_H) as a 1.0 M solution in hexanes was acquired from Sigma-Aldrich (St. Louis, MO, USA); and dichlorodimethylsilane (Me_2_SiCl_2_ ≥ 99.5%) was purchased from Sigma-Aldrich (St. Louis, MO, USA) and used as received. Cyclohexane was used as a solvent in the polymerization reactions, and it was twice distilled from sodium under an argon atmosphere before use. For abbreviation, NdV_3_ is referred to as [Nd], Al(*i*-Bu)_2_H as [Al], and Me_2_SiCl_2_ as [Cl]

### 2.1. Laboratory Homopolymerization Procedure

A preliminary study was carried out on a laboratory scale to determine the molar ratio of the catalytic system to be used in the reactors. This pre-study was performed in glass vials (50 mL) equipped with a magnetic stirrer and under a nitrogen atmosphere. The reactions were heated with an oil bath. The catalytic system was aged at room temperature for 30 min before being added to the vial glass. The monomer and the solvent were added to the vial glass, and it was heated and stirred. Then, the aged catalyst system was fed into the vial glass through a syringe to initiate the polymerization reaction and left for 4 h. The polymerization reaction was deactivated by adding acidified methanol. The result (dissolved in cyclohexane) was stabilized with Irganox 1076, precipitated in methanol, and dried under a vacuum at 25 °C until weight was constant.

### 2.2. Reactor Homopolymerization Procedure

The polymerizations were carried out under a nitrogen atmosphere in a 1L stainless steel Parr reactor equipped with a turbine-type mechanical stirrer. Heating was provided with electrical resistance and cooling with the flow of cold water through an internal tubing coil. A typical Ip polymerization procedure using NdV_3_ as catalyst, Al(*i*-Bu)_2_H as a cocatalyst, and Me_2_SiCl_2_ as an activator is described as an example: In an oven-dried, nitrogen-purged glass bottle, the catalytic system at a molar ratio of [Nd]/[Al]/[Cl] = 1.0/15/1.0 was prepared, adding the components in the following order: (i) cyclohexane as the solvent, (ii) Al(*i*-Bu)_2_H, (iii) NdV_3_, and (iv) Me_2_SiCl_2_. The catalytic system was aged at room temperature for 30 min before being added to the reactor. The monomer and cyclohexane were added to the reactor, and it was heated and stirred. Then, the aged catalyst system was fed into the reactor through a syringe to initiate the polymerization reaction. The yield percent of polymerization reactions was determined gravimetrically; to do this, samples of the reactive mixture were taken at different time intervals and the mass content of the polymer present in each sample was calculated. The yield percent was calculated using the ratio between the polymer mass content in each sample, taken at specific times, with respect to the total polymer mass content present in the reactive mixture considering 100% monomer conversion. The polymerization reaction was deactivated by adding acidified methanol; the material was stabilized with Irganox 1076, precipitated in methanol, and dried under vacuum at 25 °C until weight was constant. 

### 2.3. Characterization

The molecular weight of the samples was determined with size exclusion chromatography (SEC) using a PLGel mixed column in a PL-GPC 50 from Agilent equipped with a refractive index detector. The calibration curve was carried out with polystyrene (PS) standards, and tetrahydrofuran (HPLC-grade from Aldrich) was used as an eluent at a flow rate of 1 mL/min. The molar mass number and weight averages of the polymers, ${\bar{M}}_{n}$ and ${\bar{M}}_{w}$, respectively, relative to polystyrene standards, were corrected using the Mark–Houwink–Sakurada equation (Equation (1)):(1)$$\left[\eta \right]=K{\left(M_{v}\right)}^{\alpha }$$
where [*η*] is the intrinsic viscosity, $M_{v}$ is the viscosimetric average molar mass of the polymer, and K and α are the unique parameters for each polymer–solvent system. The Mark–Houwink–Sakurada parameters use THF as the solvent; polystyrene $\alpha_{\mathrm{PS}}=0.712$, $K_{\mathrm{PS}}=12.8\times 10^{-5}\;\mathrm{dL}/g$, polymyrcene $\alpha_{\mathrm{PMy}}=0.772$, $K_{\mathrm{PMy}}=7.46\times 10^{-5}\;\mathrm{dL}/g$ [44]; and Polyisoprene $\alpha_{\mathrm{PIp}}=0.735$, $K_{\mathrm{PIp}}=1.77\times 10^{-4}\;\mathrm{dL}/g$ [45]. The reported molecular weights in the polyfarnesene were all relative to the PS standards and not adjusted with Mark−Houwink parameters.

The microstructure of the polyisoprene (PIp), polymyrcene (PMy), and polyfarnesene (PFa) samples was calculated by ^1^H and ^13^C nuclear magnetic resonance (NMR) in a Bruker-400 MHz spectrometer. CDCl_3_ was used as a solvent and the analyses were performed at room temperature. The thermal behavior was determined with differential scanning calorimetry (DSC) using Instruments DSC 2920 equipment. The analyses were carried out under a nitrogen inert atmosphere using a heating rate of 5 °C/min. Each sample was heated twice to eliminate the thermal history.

## 3. Results and Discussion

Four conditions are necessary to achieve the CCTP regime: (i) the chain transfer reactions must be reversible; (ii) the chain transfer rate must be faster than the polymerization rate; (iii) there must be a linear dependence between the growing chains with respect to time; and (iv) chain termination reactions, such as β-hydride elimination, must be essentially negligible. This reaction scenario allows for the exhibition of three main features that define the CCTP and can be associated with the characteristics of living polymerization: (1) the average number molecular weight (${\bar{M}}_{n}$) of the polymer must have a linear relationship to the polymer yield; (2) the MWD of the produced polymer must be narrow, usually with dispersities (Ð) less than two; and (3) the number of polymer chains produced per single primary metal atom (Np) must be between 6 and 10 [29]. 

In the present paper, we evaluated the polymerization reaction conditions required to work in a CCTP regime to polymerize Ip, My, and Fa via a homogeneous ternary catalyst system composed of NdV_3_/Al(i-Bu)_2_H/Me_2_SiCl_2_. Different reaction conditions, such as [Cl]/[Nd] and [Al]/[Nd] ratio and reaction temperatures were evaluated. 

### 3.1. Effect of [Cl]/[Nd] Ratio

In terms of reaction conditions, CCTP mainly depends on the appropriate amount of chain transfer agent (CTA), which must promote faster chain transfer reactions than the growth of the chains ($k_{\mathrm{tr}}>>>k_{p}$), thus avoiding chain termination reactions. Nevertheless, the halide donor is also crucial in the catalyst activation process, where the active species are generated, and therefore, the growth of the chains may come to depend on this component (see Figure 1). Table 1 shows the results of a preliminary study (laboratory scale in 50 mL glass vials for 4 h) to determine the [Cl]/[Nd] ratio at which this system is able to work under CCTP conditions (before polymerizing in the 1L steel reactor, where the rest of the parameters were evaluated, as well as the material’s properties; it is noteworthy that only the terpenes were polymerized in glass vials for the pre-study, as isoprene was directly polymerized in a reactor), where the ${\bar{M}}_{n}$ and Ð values of the Ip, My, and Fa polymerizations were observed with conversions above 90% of the monomer. As can be observed in all cases, when the [Cl]/[Nd] ratio increased from 1.0 to 3.0, both the ${\bar{M}}_{n}$ and Ð increased, suggesting that the CCTP condition was lost and thus became a traditional coordination polymerization characterized by higher Ð (see Table 1 entries My-2 and Fa-2). In general, the use of low [Cl]/[Nd] ratios lead to single catalytic species [46]; these sub-halogenated catalytic species appear to be of lower activity than the hyperhalogenated species, as indirectly indicated by the times required to reach high conversions under the same conditions of temperature, monomer concentration ([M]), and [Al]/[Nd] ratios. In Table 2 and Table 3, both polyterpenes show lower molecular weights than PIp; this is due to the [Nd]/[M] ratio of 150 used for the polymerization of My and Fa to obtain the conversion of the monomer above 80% and Ð < 2.

According to the scheme shown in Figure 1, CCTP is catalyzed by way of neodymium complexes through a coordination mechanism where the main group metal (MGM) alkyl is used as a CTA, “surrogating” metal chain growth. During CCTP, a very fast and reversible chain transfer reaction between the active chain growth state (CGS) and the dormant chain transfer state (CTS) takes place; therefore, the chain growth is only possible through the active species belonging to the Nd catalyst. Finally, once the polymer chains are exchanged with the MGM, the chains remain at the dormant centers, at which point, the chain termination reactions are suppressed. Considering this, both ${\bar{M}}_{n}$ and Ð depend on the coordination strength between the active species of the metal (in our case, Nd) catalyst and MGM [47]. Very strong coordination can result in the growth of fewer chains with high ${\bar{M}}_{n}$ and Ð values. Thus, increasing the [Cl]/[Nd] ratio seems to improve the coordination strength of the Nd atom, which is highly possible since it contains more than one chlorine atom. The number of polymer chains that grow by each active Nd center (referred to as $N_{p}$; see Equation (1)) is indirect evidence of the higher coordination strength with specific active species that are formed at [Cl]/[Nd] ratios close to 2.0. Insoluble dihalogenated or trihalogenated species of less activity form due to excessive chlorination [48]. Similar behavior was observed by Quirk et al. [49] in 1,3-butadiene polymerization using the Ziegler–Natta NdV_3_/DIBAH/SiCl_4_ ternary catalytic system; using this catalytic system, the optimal [Cl]/[Nd] ratio was 1.0, perhaps as a result of the greater content of chloride in the SiCl_4_ with respect to Me_2_SiCl_2_. The results obtained by Quirk et al., with [Cl]/[Nd] ratios higher than 1.0, were characterized by high values of both ${\bar{M}}_{n}\;$ (>500,000 g/mol) and broad Ð (>11). It is important to mention that these polymerizations were not carried out in the CCTP regime.

This behavior is corroborated by previous results obtained by our research group, where we showed that the highest catalytic activity using the ternary system NdV_3_/Al(*i*-Bu)_2_H/Me_2_SiCl_2_ in 1,3-butadiene polymerization was reached when the [Cl]/[Nd] ratio was 2.0, and the use of higher ratios of [Cl]/[Nd] resulted in higher molecular weight polybutadienes with higher dispersity values [50].

### 3.2. Effect of [Al]/[Nd] Ratio

Table 2, Table 3 and Table 4 show the reaction conditions for the Ip, My, and Fa polymerizations, respectively. The reactions were carried out at two different temperatures (60 °C and 70 °C) and changing the [Al]/[Nd] ratios from 15 to 30. In all cases, independently of the temperature, an increase in the [Al]/[Nd] ratio led to higher polymerization rates (evaluated by gravimetrically tracking the monomer consumption) (Figure 2). On the other hand, an increase in the [Nd]/[M] ratio, from 150 to 250, with a lower [Al]/[Nd] ratio, led to an increase in the polymerization rate by decreasing the number of active sites present in the system, performing a slow insertion of the monomer into the living chain (Table 3, My-11, and Table 4, Fa-15), presumably as a result of the higher number of chain transfer reactions and the higher number of dormant species [42]. The number of polymer chains produced per Nd atom ($N_{P}$) was calculated considering the yield (%) and by means of Equation (2) [30]:(2)$$Np=\frac{{\left[M\right]}_{0}/{\left[\mathrm{Nd}\right]}_{0}\cdot M_{w\;}\cdot yield}{{\bar{M}}_{n}}$$
where [M]_0_ and [Nd]_0_ correspond to the initial concentration of the monomer and the neodymium catalyst, respectively. *M*_w_ is the molar mass of the monomer and ${\bar{M}}_{n}$ is the experimental number–average molecular weight. The Ð of the resultant polymers, based on the three monomers, indicates that this behavior does not depend only on the [Al]/[Nd] molar ratio, but also on the steric volume of the monomers, observing that the lowest Ð were obtained with the less bulky monomer (Ip). Additionally, when the polymerization temperature was increased from 60 to 70 °C, the Ð decreased, suggesting the polymerization rate’s controlling effect over dispersion when CCTP conditions prevailed; that is, the propagation rate increased while the chain transfer reactions did not show significant influence on the overall polymerization rate, as demonstrated by our results and also previously observed by Coutinho et al. [51]. Additionally, the same effect was observed for the Fa polymerization with monomer/initiator ([M]/[Nd]) ratios of 250 at different polymerization temperatures (Table 4, entries PFa-11 to PFa-14) where one can observe a Ð decrease reaching a constant value near to 1.6 when the temperature polymerization was increased from 50 to 80 °C.

The effect of the [Al]/[Nd] ratio over the microstructure was further studied, and structural isomers 3,4 and 1,4 (*cis* + *trans*) were determined with ^1^H NMR spectra, integrating signals in the region of the olefinic groups located in a range from 4.7 to 5.3 ppm. The *cis*/*trans* ratio was calculated with the ^13^C NMR spectra (proton-gated decoupling no-NOE experiments) [48] by also integrating the signals of olefinic groups. Figure 3 and Figure 4 show the ^1^H and ^13^C NMR spectra and signal assignations for an example of each of the synthesized polymers. In general terms, an increase in the [Al]/[Nd] ratio led to a decrease in the 1,4-*cis* content in the polymerization of the three monomers. This behavior suggests that a higher number of reversible chain transfer reactions between the dormant species and active species could interrupt the sequential *cis* linkage, giving rise to occasional *trans* links during the reversible chain transfer reactions. Finally, according to the observed glass transition temperature (*T_g_*), the two synthesized polyterpenes can be considered amorphous materials with rubbery characteristics.

### 3.3. Effect of Temperature

In order to validate the CCTP conditions, i.e., that the reversible chain transfer is faster than the polymerization rate, kinetic studies were carried out for the polymerization of Ip, My, and Fa at 60 °C and 70 °C. The propagation ($k_{p}$) and chain transfer ($k_{\mathrm{tr}}$) constants were determined. The propagation rate constant, or $k_{p}$, was calculated with Equation (3) [49], which describes the polymerization rate ($R_{p}$) and the corresponding plots as a function of time (Equation (4)), as shown in Figure 5.
(3)$$R_{P}=-\frac{d\left[M\right]}{\mathrm{dt}}=k_{p}\left[M\right]{\left[\mathrm{Nd}\right]}_{0}$$
(4)$$\frac{-\ln\left(1-x\right)}{t}=k_{p}{\left[\mathrm{Nd}\right]}_{0}$$
where *x* is the conversion degree and [Nd]_0_ is the initial concentration of the neodymium catalyst. For each of the different systems (Ip, My, and Fa polymerizations), the conversion degree increased as a function of time, and −ln (1 − *x*) was proportional to the polymerization time in a linear behavior. This indicated that the polymerization rate was first order with respect to the monomer concentration, displaying the living nature of the polymerization reactions. In this sense, Equation (2) is valid to determine $k_{p}$. The chain transfer rate constant, or $k_{\mathrm{tr}}$, was estimated by solving the function relation between the polymerization degree (PD) and polymerization time expressed by Equation (5) in order to obtain Equation (6) [31].
(5)$$PD=\int_{0}^{t}\frac{1}{k_{\mathrm{tr}}{\left[\mathrm{Nd}\right]}_{0}{\left[\mathrm{Al}\right]}_{0}}\left(e^{\ln{\left[M\right]}_{0}}-e^{\ln{\left[M\right]}_{0}-k_{p}{\left[\mathrm{Nd}\right]}_{0}t}\right)\mathrm{dt}$$
(6)$$PD=\frac{1}{k_{\mathrm{tr}}{\left[\mathrm{Nd}\right]}_{0}{\left[\mathrm{Al}\right]}_{0}}\left\{{\left[M\right]}_{0}t+\frac{1}{k_{p}{\left[\mathrm{Nd}\right]}_{0}}\left(e^{\ln{\left[M\right]}_{0}-k_{p}{\left[\mathrm{Nd}\right]}_{0}t}-{\left[M\right]}_{0}\right)\right\}$$
where [Nd]_0_, [Al]_0_, and [M]_0_ are the initial concentration of the neodymium catalyst, Al(*i*-Bu)_2_H, and monomer, respectively; $k_{p}$ is the rate constant of propagation; and t is the polymerization time. The results, shown in Figure 5, demonstrate that the $k_{\mathrm{tr}}$ was 200–300 times higher than the $k_{p}$ when the [Al]/[Nd] and [Cl]/[Nd] ratios were 25 and 1.0, respectively. From this, we can conclude that, under such specific conditions, it is possible to carry out the polymerization of terpenic monomers under a CCTP regime. These conditions significantly differ in terms of [Cl]/[Nd] ratios compared to those reported by other authors for Ip and 1,3-butadiene using the catalytic systems NdV_3_/Al(*i*-Bu)_2_H/Me_2_SiCl_2_ and Nd(OiPr)_3_/Al(*i*-Bu)_2_H/Me_2_SiCl_2_ [31]. Moreover, the activation energy for the corresponding propagation and chain transfer processes was determined by taking into account the Arrhenius equation (Equation (7)): (7)$$k=f\left(T\right)=Ae^{\frac{-E}{RT}}\;$$
where *T* is the absolute temperature, E is the corresponding activation energy, and *R* is the universal gas constant (= 1.98 cal/K mol) (Table 5).

### 3.4. Validation of CCTP Regime through Living Polymerizations (Sequential Polymerizations)

As mentioned above, the absence of chain-termination reactions and the linear dependence of chain growth as a function of the polymerization time represent two of the four conditions that define CCTP and are, at the same time, characteristics of living polymerization. In this scenario, the polymerization of the monomers Ip, My, and Fa using the catalytic system [NdV_3_]/[Al(*i*-Bu)_2_H]/[Me_2_SiCl_2_] under a CCTP regime must allow the sequential addition of monomer to the reactive system; that is, to feed an initial amount of monomer (first step), and when it has reached 100% of yield, to feed the second amount of monomer (second step), giving rise to an increase in the molecular weight of the polymer without significantly affecting its dispersion. According to these criteria, the catalytic [NdV_3_]/[Al(*i*-Bu)_2_H]/[Me_2_SiCl_2_] should carry out the living polymerization of the monomers under study, keeping the number of active species ($N_{P}$) constant, which indicates that the concentration of the living polymer chains must remain unchanged during the global polymerization reaction [31]; additionally, the chain transfer kinetics constant ($k_{\mathrm{tr}}$) should be higher than the propagation kinetics constant ($k_{p}$) in both steps. In order to validate the CCTP condition regime in our experiments, sequential polymerizations of the monomers Ip, My, and Fa were carried out, and the results of both steps are shown in Table 6.

In Figure 6, it can be seen that the molecular weight increases with the yield (reaction time), while the Ɖ values remain relatively constant, and this behavior prevails in the second stage of the polymerization reaction for the three monomers. The above results demonstrate the living state of the polymerizations carried out and the CCTP regime of the same. It is important to mention that, at the beginning of the polymerization reactions (polymer yield near 20–30%), the values of Ɖ are considerably high (Ip-11: Ð = 2.2, My-11: Ð = 2.5, and Fa-15: Ð = 4); however, these gradually decrease by increasing the polymer yield until reaching constant values when the yields reach values higher than 50–60% (Figure 7). This behavior can be explained as a result of an induction time during chain initiation promoted by the steric volume of the monomers, resulting in the initial values of Ɖ being greater in the polymerization of a monomer of greater steric volume (Fa).

In addition to the results mentioned above (Figure 6 and Figure 7), in Table 6, we show the behavior of ${\bar{M}}_{n}$ and $N_{P}$ with respect to the reaction time, and we can observe that the number of active species ($N_{P}$) reaches a stable state while the ${\bar{M}}_{n}$ of the synthesized polymers continues to increase as much in the first as in the second stage of polymerizations. This behavior also demonstrates the living features of the polymerizations carried out, since the phenomenon of deactivation in the active species was not observed at the moment of adding the second monomer load (second step).

Finally, in the sequential polymerizations, it was possible to calculate the chain transfer kinetics constant ($k_{\mathrm{tr}}$) and the propagation kinetics constant ($k_{p}$) for both stages in the polymerization of all monomers (Ip, My, Fa) (Table 6) using Equation 3 and 6. In all cases, the $k_{\mathrm{tr}}>>>k_{p}$. These results, together with the living features of the polymerizations and the partially constant number of active species ($N_{P}$) in both stages of the sequential polymerizations demonstrate that, in our experiment, it was possible to work in a CCTP regime.

### 3.5. Nd-Catalyst: Efficiency and Economy in CCTP 

The CCTP process allows us to obtain a high catalyst efficiency since one Nd atom can generate more than one polymer chain depending on the reaction conditions and molar ratios ([M]/[Nd], [Al]/[Nd], and [Cl]/Nd]) compared to the conventional living polymerization. Furthermore, the quantity of the polymer product is not determined by the amount of Nd-catalyst, but rather by the concentration of the less expensive Al(*i*-Bu)_2_H or CTA. For this reason, and according to Georges et al. [39], a significant catalyst economy is thus possible through CCTP. In our case, Figure 8 shows the behavior of $N_{P}$ and ${\bar{M}}_{n}$ as a function of the [Al]/[Nd] molar ratio (at complete polymerization), and it can be seen that the $N_{P}$ increases in practically all cases, almost linearly, from 2 to 8, increasing the [Al]/[Nd] molar ratio from 15 to 35, indicating the remarkable influence of the aluminum concentration over the reversible chain transfer and, therefore, $N_{P}$. Additionally, ${\bar{M}}_{n}$ tends to decrease as the [Al]/[Nd] molar ratio increases. Considering the dependence of $N_{P}$ on the [Al]/[Nd] molar ratio from Figure 8, the ${\bar{M}}_{n}$ was established, resulting in Equations (8)–(10). Considering the above, CCTP represents an excellent path of polymerization toward the creation of an efficient and high atom–neodymium economy, polymerizing biobased monomers to access sustainable rubbers with controlled microstructures and macrostructures.
(8)$${\bar{M}}_{n}=\frac{\left[M\right]/\left[\mathrm{Nd}\right]\cdot M_{\mathrm{Ip}}\cdot yield}{0.4696\left(\left[Al\right]/Nd]\right)-4.2985}$$
(9)$${\bar{M}}_{n}=\frac{\left[M\right]/\left[\mathrm{Nd}\right]\cdot M_{\mathrm{My}}\cdot yield}{0.3601\left(\left[Al\right]/Nd]\right)-2.1946}$$
(10)$${\bar{M}}_{n}=\frac{\left[M\right]/\left[\mathrm{Nd}\right]\cdot M_{\mathrm{Fa}}\cdot yield}{0.3344\left(\left[Al\right]/Nd]\right)-1.6464}$$

## 4. Conclusions

The biobased terpene monomers myrcene and farnesene were polymerized via coordination chain transfer polymerization (CCTP) by using NdV_3_/Al(*i*-Bu)_2_H/Me_2_SiCl_2_, taking isoprene as a reference monomer. The catalyst system proved to be highly efficient for all monomers, resulting in polymer yields greater than 85% in a short time (1 h maximum) using a low [Cl]/[Nd] molar ratio (=1) and different [Al]/[Nd] molar ratios, from 15 to 30. The dispersity (Ð) via CCTP depended on the [Al]/[Nd] molar ratio, in addition to the steric volume of the monomers, while observing the lowest Ð with the less bulky monomer (isoprene). The kinetic results show that the chain transfer ($k_{\mathrm{tr}}$) was 200–300 times higher than the propagation constant ($k_{p}$), which is consistent for a CCTP process. Additionally, two-step polymerizations were performed where the living nature of CCTP was evidenced. Finally, the results of the polyterpenes we obtained show that they are a good option for a partial substituent of isoprene, as they present similar characteristics for their polymerization and chemical properties.

## Figures and Tables

**Figure 1 polymers-14-02907-f001:**
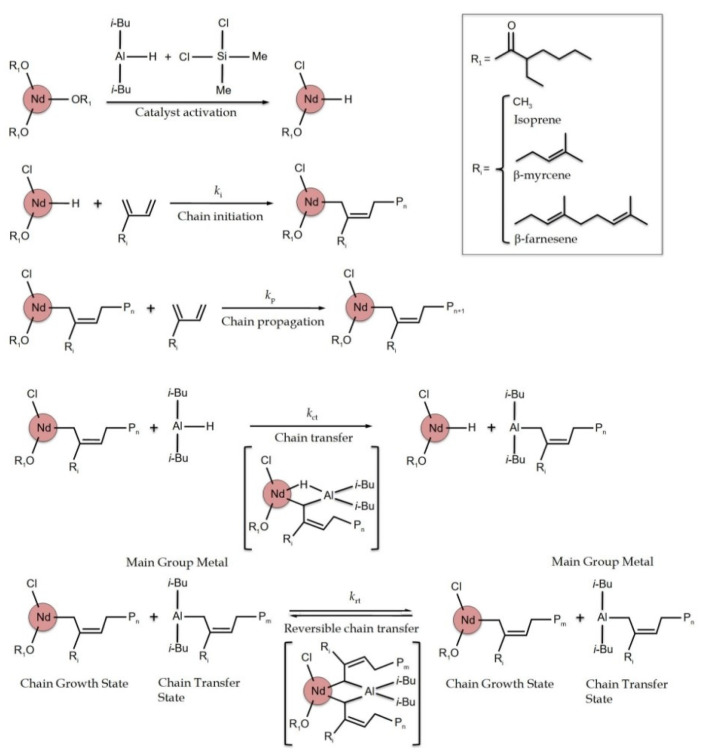
Reversible coordinative chain transfer polymerization mechanism applied for 1,3-diene initiated by an NdV_3_/Al(*i*-Bu)_2_H/Me_2_SiCl_2_ catalyst system. Adapted from references [29,30,42,43].

**Figure 2 polymers-14-02907-f002:**
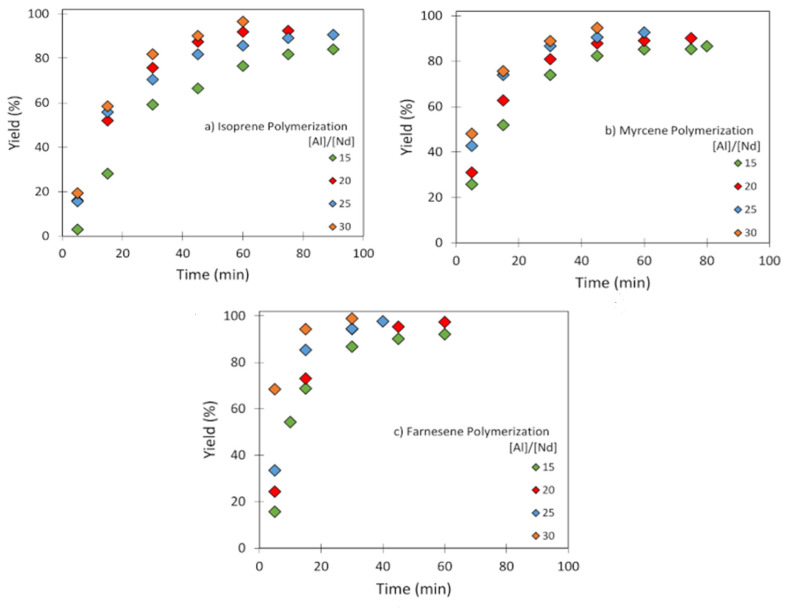
Conversion evolution as a function of time using different [Al]/[Nd] ratios for (**a**) Ip: entries Ip-3 to Ip-6; (**b**) My: entries My-3 to My-6; (**c**) Fa: entries Fa-3 to Fa-6, polymerizations initiated by NdV_3_/Al(*i*-Bu)_2_H/Me_2_SiCl_2_ in cyclohexane at 60 °C.

**Figure 3 polymers-14-02907-f003:**
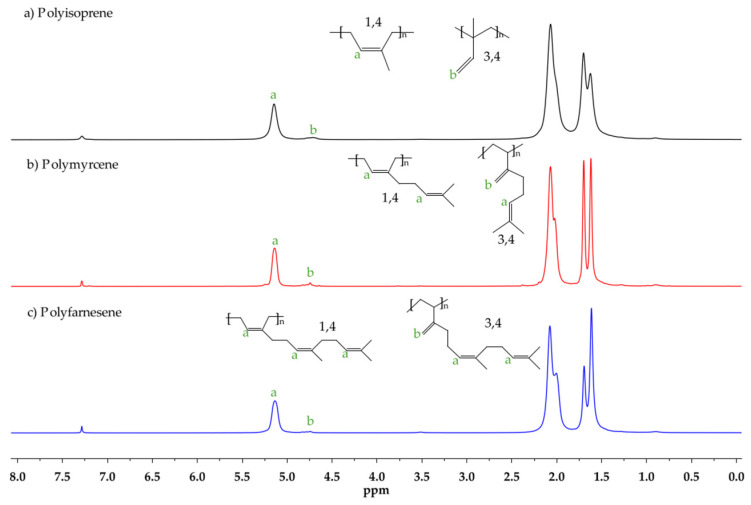
^1^H NMR spectra using CDCl_3_ as solvent on (**a**) Ip: entry Ip-11, (**b**) My: entry My-11, and (**c**) Fa: entry Fa-15.

**Figure 4 polymers-14-02907-f004:**
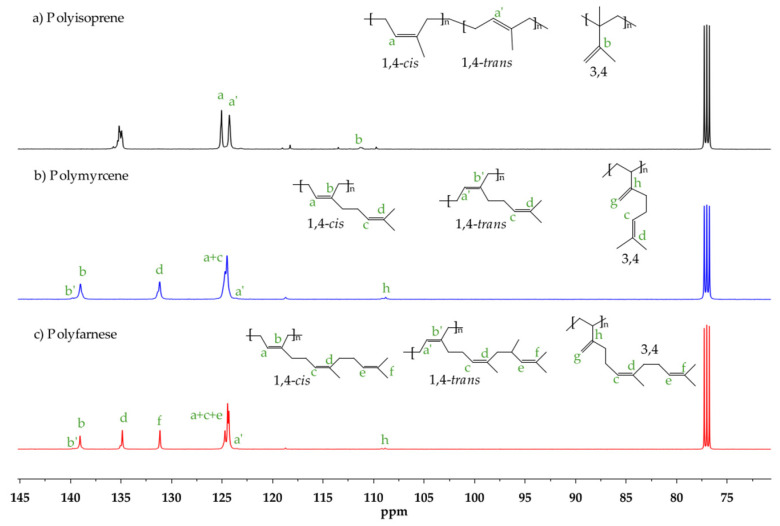
^13^C NMR spectra using CDCl_3_ as a solvent on (**a**) Ip: entry Ip-11, (**b**) My: entry My-11, and (**c**) Fa: entry Fa-15.

**Figure 5 polymers-14-02907-f005:**
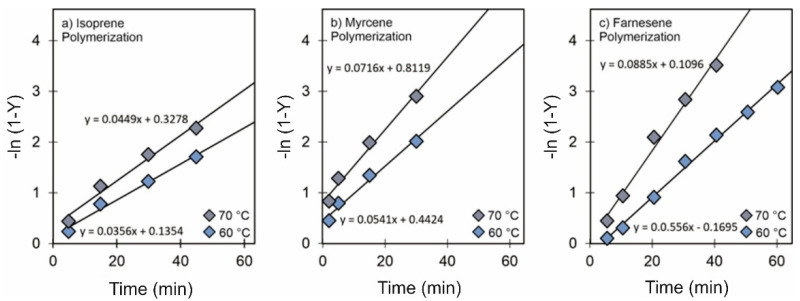
Effect of temperature on the reaction rate for (**a**) Ip: entries Ip-5 and Ip9, (**b**) My: entries My-5 and My-9 and (**c**) Fa: entries Fa-12 and Fa-13 polymerizations initiated by NdV_3_/Al(*i*-Bu)_2_H/Me_2_SiCl_2_ in cyclohexane.

**Figure 6 polymers-14-02907-f006:**
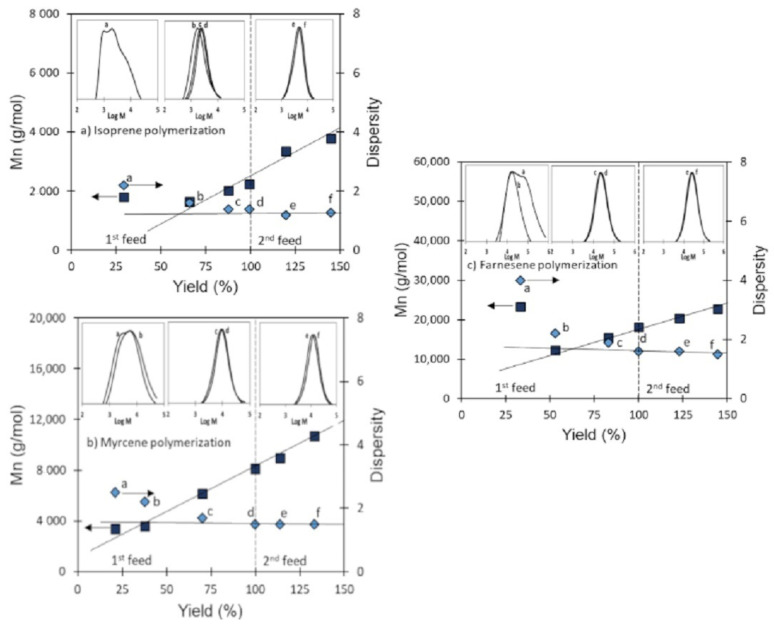
Dependence of ${\bar{M}}_{n}$ and dispersity against yield in (**a**) Ip: entry Ip-11, (**b**) My: entry My-11, and (**c**) Fa: entry Fa-15, polymerizations initiated by NdV_3_/Al(*i*-Bu)_2_H/Me_2_SiCl_2_ in cyclohexane at 70 °C.

**Figure 7 polymers-14-02907-f007:**
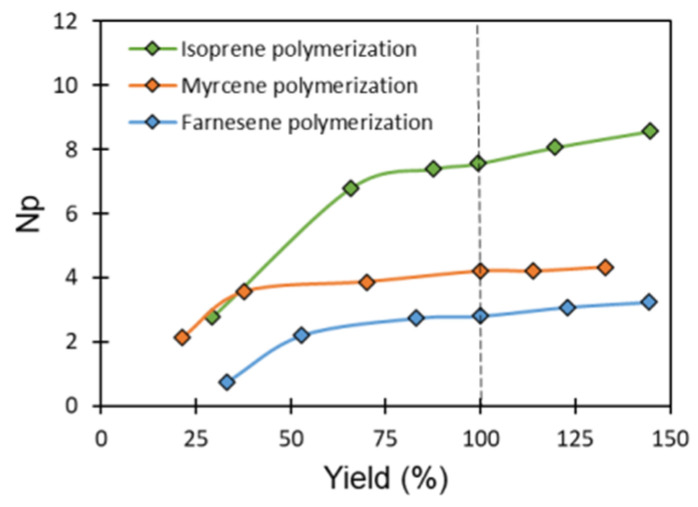
Dependence of $N_{P}$ against yield for isoprene, myrcene and farnesene polymerizations initiated by NdV_3_/Al(*i*-Bu)_2_H/Me_2_SiCl_2_ in cyclohexane at 70 °C.

**Figure 8 polymers-14-02907-f008:**
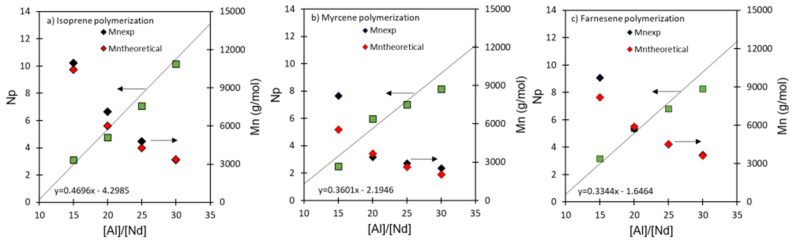
${\bar{M}}_{n}$, ${\bar{M}}_{\mathrm{ntheoretical}}$, and $N_{P}$ as functions of [Al]/[Nd] ratios for (**a**) Ip: entries Ip-3 to Ip-6, (**b**) My: entries My-3 to My-6, and (**c**) Fa: entries Fa-3 to Fa-6, polymerizations initiated by NdV_3_/Al(*i*-Bu)_2_H/Me_2_SiCl_2_ in cyclohexane at 60 °C.

**Table 1 polymers-14-02907-t001:** Polymerizations initiated by NdV_3_/Al(i-Bu)_2_H/Me_2_SiCl_2_ in cyclohexane at 70 °C.

Entry	[Nd]:[Al]:[Cl]	[Nd]_0_ (μmol)	Time (min)	Yield ^a^ (%)	${{\bar{M}}_{n}}^{b}\;(g/\mathrm{mol})$	Ð ^b^	${N_{P}}^{c}$
**Ip-1 ^d^**	1:20:01	1.73	75	99	2600	1.3	2.7
**Ip-2 ^d^**	1:20:03	1.73	45	93	8000	1.8	2.3
**My-1 ^e^**	1:20:01	0.15	240	96	2900	1.6	6.1
**My-2 ^e^**	1:20:03	0.15	240	96	8300	3	2.5
**Fa-1 ^e^**	1:20:01	0.09	240	99	7600	2.3	4
**Fa-2 ^e^**	1:20:03	0.09	240	99	24,000	4.1	1.3

Ip: isoprene; My: myrcene; Fa: farnesene. ^a^ The polymer yield was calculated gravimetrically based on the monomer feed. ^b^ Determined with size exclusion chromatography using polystyrene standards at 25 °C in THF as an eluent. ^c^ Average number of polymer chains produced by a single Nd atom ($N_{P}$) determined according to Equation (2). ^d^ Reaction conditions: cyclohexane = 130 mL, Ip = 26 mL; Temp rxn: 70 °C. ^e^ Reaction conditions: cyclohexane = 18 mL, My or Fa = 3.5 mL; polymerizations were carried out under a nitrogen atmosphere in a 50 mL vial glass. Temp rxn: 70 °C.

**Table 2 polymers-14-02907-t002:** Isoprene polymerization initiated by NdV_3_/Al(*i*-Bu)_2_H/Me_2_SiCl_2_ in cyclohexane at different reaction conditions.

Entry ^a^	[Nd]:[Al]:[Cl]	T (°C)	Time (min)	Yield ^b^ (%)	${{\bar{M}}_{n}}^{c}\;(g/\mathrm{mol})$	Ð ^c^	${N_{P}}^{d}$	Isoprene Structure (%) ^e^			${T_{g}}^{f}$ (°C)
								3,4	1,4-*trans*	1,4-*cis*	
**Ip-3**	1:15:01	60	90	84	10,900	1.8	2.6	3	15.5	81.5	−68.2
**Ip-4**	1:20:01	60	90	90	7100	1.5	4.1	3	22.3	74.7	−69.4
**Ip-5**	1:25:01	60	75	92	4800	1.4	6.6	3.2	27.5	69.3	−69.2
**Ip-6**	1:30:01	60	60	97	3300	1.4	9.9	3	27	70	−70.5
**Ip-7**	1:15:01	70	80	87	9700	1.5	3.2	3	26.5	70.5	−66.4
**Ip-8**	1:20:01	70	80	93	6400	1.4	4.9	4	32	64	−68.3
**Ip-9**	1:25:01	70	70	94	3800	1.4	8.2	3	34	63	−69.3
**Ip-10**	1:30:01	70	55	96	3600	1.3	8.2	3	32	65	−70.3

^a^ [Nd]_0_ = 6.4 × 10^−4^ mol, [Ip]/[Nd] = 500, cyclohexane = 140 mL, and Ip = 32 mL. ^b^ Polymer yield calculated gravimetrically based on the monomer consumption. ^c^ Determined with SEC using polystyrene standards at 25 °C in THF as an eluent. ^d^ Average number of polymer chains produced by a single Nd atom ($N_{P}$) determined according to Equation (2). ^e^ Ip structures determined with ^1^H NMR and ^13^C NMR. ^f^ Glass transition temperature determined with DSC at 5 °C/min.

**Table 3 polymers-14-02907-t003:** Myrcene polymerization initiated by NdV_3_/Al(*i*-Bu)_2_H/Me_2_SiCl_2_ in cyclohexane under different reaction conditions.

Entry ^a^	[Nd]:[Al]:[Cl]	T (°C)	Time (min)	Yield ^b^ (%)	${{\bar{M}}_{n}}^{c}\;(g/\mathrm{mol})$	Ð ^c^	${N_{P}}^{d}$	β-myrcene Structure (%) ^e^			${T_{g}}^{f}$ (°C)
								3,4	1,4-*trans*	1,4-*cis*	
**My-3**	1:15:1	60	80	85	8200	1.5	2.2	6.3	3.4	90.3	−68.4
**My-4**	1:20:1	60	75	90	4000	1.5	4.6	6.3	1.6	92.1	−72.2
**My-5**	1:25:1	60	60	88	2900	1.8	7.3	6.1	5.6	87.4	−73.7
**My-6**	1:30:1	60	45	95	2500	1.8	6.9	9.7	16.8	73.5	−78.0
**My-7**	1:15:1	70	75	86	7100	1.3	2.5	13.0	4.1	82.9	−69.4
**My-8**	1:20:1	70	65	89	3400	1.5	5.2	8.0	2.3	89.7	−72.1
**My-9**	1:25:1	70	60	93	2500	1.6	7.6	8.1	8.7	83.2	−74.3
**My-10**	1:30:1	70	45	94	2300	1.5	7.5	10.0	18.8	71.2	−77.0

^a^ [Nd]_0_ = 1.01 × 10^−3^ mol, [My]/[Nd] = 150, cyclohexane = 130 mL, and My = 26 mL. ^b^ Polymer yield calculated gravimetrically based on the monomer consumption. ^c^ Determined with SEC using polystyrene standards at 25 °C in THF as an eluent. ^d^ Average number of polymer chains produced by a single Nd atom ($N_{P}$) determined according to Equation (2). ^e^ My structures determined with ^1^H NMR and ^13^C NMR. ^f^ Glass transition temperature determined with DSC at 5 °C/min.

**Table 4 polymers-14-02907-t004:** β-farnesene polymerization initiated by NdV_3_/Al(*i*-Bu)_2_H/Me_2_SiCl_2_ in cyclohexane at different reaction conditions.

Entry ^a^	[Nd]:[Al]:[Cl]	T (°C)	Time (min)	Yield ^b^ (%)	${{\bar{M}}_{n}}^{c}\;(g/\mathrm{mol})$	Ð ^c^	${N_{P}}^{d}$	β-farnesene Structure (%) ^e^			${T_{g}}^{f}$ (°C)
								3,4	1,4-*trans*	1,4-*cis*	
**Fa-3**	1:15:1	60	60	92	9700	2.9	3.2	9.4	1.2	89.4	−75.1
**Fa-4**	1:20:1	60	60	97	5700	2.4	5.4	3.1	2.2	94.7	−76.6
**Fa-5**	1:25:1	60	40	99	4500	2.0	6.8	3.4	4.7	91.8	−76.9
**Fa-6**	1:30:1	60	30	99	3700	2.0	8.3	4.7	8.5	86.8	−78.4
**Fa-7**	1:15:1	70	40	99	8900	2.0	3.4	4.0	1.0	95.0	−75.2
**Fa-8**	1:20:1	70	40	99	6500	2.1	4.7	3.0	1.5	95.5	−77.6
**Fa-9**	1:25:1	70	30	99	4800	1.9	6.4	3.8	7.2	89.0	−81.6
**Fa-10**	1:30:1	70	30	99	3800	2.6	8.0	4.8	5.2	90.0	−76.9
**Fa-11 ^g^**	1:25:1	50	85	92	10,500	2.5	4.9	-	-	-	-
**Fa-12 ^g^**	1:25:1	60	70	96	8200	1.8	6.2	-	-	-	-
**Fa-13 ^g^**	1:25:1	70	60	97	7800	1.7	6.6	-	-	-	-
**Fa-14 ^g^**	1:25:1	80	60	85	7100	1.6	7.2	-	-	-	-

^a^ [Nd]_0_ = 6.9 × 10^−4^ mol, [Fa]/[Nd] = 150, cyclohexane = 130 mL, and Fa = 26 mL. ^b^ Polymer yield calculated gravimetrically based on the monomer consumption. ^c^ Determined with SEC using polystyrene standards at 25 °C in THF as an eluent. ^d^ Average number of polymer chains produced by a single Nd atom ($N_{P}$) determined according to Equation (2). ^e^ Fa structures determined with ^1^H NMR and ^13^C NMR. ^f^ Glass transition temperature determined with DSC at 5 °C/min. ^g^ [Nd]_0_ = 4.9 × 10^−4^ mol, [Fa]/[Nd] = 250, cyclohexane = 155 mL, Fa = 31 mL.

**Table 5 polymers-14-02907-t005:** Effect of temperature on the propagation and chain transfer reactions.

Entry	T (K)	${k_{p}}^{a}$ (L/mol min)	${E_{a\;propagation}}^{b}\;(\mathrm{kcal}/\mathrm{mol})$	${k_{tr}}^{c}$ (L/mol min)	${E_{a\;transfer}}^{d}\;(\mathrm{kcal}/\mathrm{mol})$
**Ip-5**	333	7.6	5.2	4400	3.2
**Ip-9**	343	9.6		9060	
**My-5**	333	8.4	6.3	2005	4.6
**My-9**	343	11.1		2197	
**Fa-12**	333	20.5	10.5	3554	3.7
**Fa-13**	343	33.3		4188	

^a^ Propagation rate constant calculated from Equations (3) and (4). ^b^ Activation energy for propagation process using the corresponding $k_{p}$ at different temperatures determined according to Equation (3). ^c^ Chain transfer rate constant obtained by solving the integral in Equation (6) and using the corresponding PD and t. ^d^ Activation energy for chain transfer process using the corresponding $k_{\mathrm{tr}}$ at different temperatures determined according to Equation (7).

**Table 6 polymers-14-02907-t006:** Validation of CCTP regime through living polymerizations.

	First Step						
Entry	Time (min)	Yield (%)	Đ ^a^	${\bar{M}}_{n}{\;}^{a}$ (g/mol)	$k_{tr}$ (L/mol-min)	$k_{p}$ (L/mol-min)	$N_{P}{\;}^{b}$
**Ip-11**	30	99	1.4	2600	36,700	37.2	7.5
**My-11**	95	99	1.5	8000	9970	21.6	4.1
**Fa-15**	95	99	1.6	17,900	8150	39	2.8
	**Second Step**						
**Entry**	**Time** **(min)**	**Yield** **(%)**	**Đ ^a^**	${\bar{M}}_{n}$ ** ^a^ ** **(g/mol)**	$k_{tr}$ **(L/mol-min)**	$k_{p}$ **(L/mol-min)**	$N_{P}$ ** ^b^ **
**Ip-11**	80	144	1.3	3800	9080	6.0	8.5
**My-11**	150	132	1.5	10,700	2300	5.1	4.5
**Fa-15**	135	144	1.5	22,200	3710	16.4	3.3

^a^ Determined with SEC using polystyrene standards at 25 °C in THF as an eluent. ^b^ Average number of polymer chains produced by a single Nd atom ($N_{P}$) determined according to Equation (2). [M]/[Nd] = 250 was used for all polymerizations. Ip: [Nd]_0_ = 2.4 × 10^−3^ mol, cyclohexane = 130 mL, Ip = 60 mL (first feed), and Ip = 30 mL (second feed). My: [Nd]_0_ = 8.25 × 10^−4^ mol, cyclohexane = 143 mL, My = 36 mL (first feed), and My = 16 mL (second feed). Fa:[Nd]_0_ = 6.4 x10^−4^ mol, cyclohexane = 130 mL, Fa = 40 mL (first feed), and Fa = 20 mL (second feed).

## Data Availability

Not applicable.

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
