# Peer review of "Coordinative Chain Transfer Polymerization of Sustainable Terpene Monomers Using a Neodymium-Based Catalyst System"

_polymers, 2022, doi:10.3390/polym14142907_

Round 1
Reviewer 1 Report
This manuscript concerns the coordinative chain transfer polymerization (CCTP) of myrcene (My) and farnesene (Fa) with the ternary Ziegler-Natta catalyst system comprising [NdV3]/[Al(i-Bu)2H]/[Me2SiCl2] to obtain sustainable polymers in the presence of large molar ratio of Al(i-Bu)2H to NdV3. There are some questions should be issued before it can be accepted for publication on Polymers.
1. P2L88:”….for the Is, My and Fa polymerizations…” should be “….for the Ip, My and Fa polymerizations…”.
2. Fig.5: The -ln (1-x) was proportional to the polymerization time in a linear behavior according to equation (3). This just indicates that the polymerization rate is first order with respect to the monomer concentration if the linear relationship passed through the origin. The first order kinetics can not display the living nature of the polymerization reactions. Why the linear relationship between -ln (1-x) and polymerization time did not pass through the origin for the polymerizations of Ip, My and Fa in Fig.5? Please give the explanation.
3. Fig. 6 & Fig.7, L399-403: Both the molecular weights of the resulting polymers and the number of polymer chains produced per Nd atom (Np) increased with increasing polymer yield in the first polymerization stage. The authors should compare the measured molecular weights of the resulting polymers with the theoretical molecular weights both in the first polymerization stage and in second polymerization stage. Moreover, the authors should also give the Np data for the second polymerization stage in Fig.7. The living feature of the polymerizations should be further confirmed.
4. Fig. 8, equations (7), (8), and (9): The theoretical curves according to equations (7), (8), and (9) should be given in Fig.8.
5. The relatively low molecular weight polymers could be only synthesized in the CCTP process due to the extremely high chain transfer rate during the polymerization in this research work. Therefore, it is still far to produce sustainable rubbers with relatively high molecular weights. The authors should revise the conclusion.
6. The authors should check and revise the references in this manuscript. There are some repeated references, e.g. ref.30 is the same as ref.49; ref.31 is the same as ref.52; ref.32 is the same as ref.53.
Reviewer 2 Report
The Coordination Chain Transfer Polymerization, CCTP, of several terpenoid monomers was studied in this studied. The authors examined the experimental conditions in order to verify that the polymerization reaction proceeds under the conditions of CCTP and not as a conventional coordination polymerization. The overall quality of this work is very good and the results will certainly be beneficial to the scientific community. There are a few points that have to be taken into account by the authors before the publication of this manuscript:
· The authors have to polish their English throughout the text to improve their expressions.
· End of Introduction: “The living nature of the system was evaluated…” Certainly the system is not living at all. The expressions should be changed to: “The controlled nature of the system was evaluated…”.
· The molecular weights of the samples were calculated by SEC using polystyrene standards and the corresponding calibration curve. These data were further employed for the calculation of other parameters (such as Np). It is known that the results of SEC analysis provide only apparent molecular weights and that the deviation from the real molecular weights is significant I propose the authors to provide real molecular weights and re-calculated the parameters that are affected from the Mn values.
· Since re-initiation experiments are efficient is it possible to prepare well-defined block copolymers from the monomers employed in this study?
Author Response
Please see the attachment

This manuscript is a resubmission of an earlier submission. The following is a list of the peer review reports and author responses from that submission.
Round 1
Reviewer 1 Report
This manuscript concerns the coordinative chain transfer polymerization of bio-based monomers with a ternary neodymium catalyst system NdV3/Al(i-Bu)2H/Me2SiCl2 to prepare the sustainable rubbers. There are some questions should be issued before it can be accepted for publication on Polymers.
- The title is quite similar to those in the published papers. It should be revised on the basis of the results in this investigation.
- The NdV3/Al(i-Bu)2H/Me2SiCl2 ternary catalyst system has been used in coordination polymerizations of conjugated dienes. The related references should be cited in this manuscript.
- The first report of coordinative chain transfer polymerization and the proposed mechanism should be described in detail and the related references should also be cited in this manuscript.
- Figure 1 gives the reversible coordinative chain transfer polymerization mechanism, which has been similarly reported in reference.
- The polymers obtained by the coordinative chain transfer polymerization in this manuscript have relatively low molecular weights. How to increase the molecular weight of the resulting polymers to be high enough as rubber materials?
- The authors said that the catalyst system exhibits a very high catalytic efficiency. What is the dosage of catalyst used to obtain close to 100% of monomer conversion? It should be recognized that the catalyst system does not exhibit very high catalytic efficiency if the molar ratio of monomer to catalyst is relatively low.
Reviewer 2 Report
In this paper, Diaz de Leon and coworkers reported on the Coordinative Chain Transfer Polymerization (CCTP) of isoprene and other bio-based terpenes, namely ß-myrcene and trans-ß-farnesene, promoted by the catalytic system NdV3/Al(iBu)2H/Me2SiCl2. They obtained polyterpenes with narrow molecular weight distributions, and with a considerable amount of cis-1,4 unit concerning the microstructure.
I am aware of the effort that writing a research article requires, and for this, I am sorry for the final decision of rejecting this paper. However, at this point, the article requires a full revision in both the form and the content.
Here I highlight the main issues that I strongly suggest considering before resubmitting the article.
- First of all, extensive editing of the English language and style is required. In many sections of the article, incorrect use of the English language prevents a full understanding of the text and of the results reported. Moreover, many grammatical mistakes can be encountered along with the text and must be corrected.
- Title: mentioning the word "rubber" and not reporting the mechanical tests of the obtained sample is inappropriate. The title should be changed to something less epic but more related to the topic investigated in the research activity presented.
- In the Introductory paragraph, among the polymerization paths for the synthesis of polyterpenes, is not mentioned the vinyl addition polymerization, which indeed gives interesting results in terms of monomer conversion, polymer molecular weight, and in some cases also mechanical properties. For this reason, in order to give the reader a more comprehensive overview of the terpenes polymerizations, I suggest adding some interesting papers (as an example Macromolecules 2020, 53, 1665-1673).
- I did not understand the point concerning the gravimetric evaluation of the polymerizations (lines 124-126). Later in the text, this part is not described, and no partial results are reported in the tables. More details should be added to better understand the use of these data.
- Ref. 39 is invoked to corroborate the results that correlate the [Cl]/[Nd] molar ratio and Mn and D. However, (a) ref. 39 is a conference proceeding and not a peer-reviewed paper, and for this, a more solid reference should be added; (b) in lines 196-200 it is mentioned the polymerization of 1,3-butadiene, while the title of ref. 39 is about the polymerization of ß-myrcene.
- Figure 1: a problem occurred in transferring the images since a "?" is reported in each line.
- Table 1: the data reported in this table should be more detailed in the main text. I have some questions in particular: (1) what is the rationale behind the choice of the different reaction times? (2) Since the catalytic system described in the text is reported as being living, can you give some results of the increased yield and molecular weight with time, other than the results reported in Figure 2? (For which I have a comment on that later on) (3) Can you give a comment on the low molecular weights obtained?
- Line 238: 3,4 and 1,4 units are structural isomers rather than geometrical isomers (which instead are cis-1,4 and trans-1,4 units).
- Lines 240-243: which is the reference article used for the calculation of the cis/trans ratio?
- Table 3: what is the rationale behind the choice of the different reaction times?
- Table 4: can you give a comment on the "N.D." reported? What's the problem with microstructure analysis or Tg determination?
- Figure 2 reports some graphs concerning the yield evolution along time, as a function of the [Al]/[Nd] ratio. Is it properly employed the parameter "yield"? I would rather use the monomer conversion if these data are taken from the gravimetric evaluation along time. In addition, all the graphs interrupt at 60 mins, at which almost all polymerizations are reported to reach 100% of yield; however, there are many entries in the previous tables that have polymerization times longer than 60 mins (entries Ip-3 and Ip-4 t=90'; Ip-5 t=75'; My-3 t=80'; My-4 t=75') and that do not reach full conversion of the monomers and these times. For this, I believe that Figure 2 is misleading in reporting the data obtained and should be completely rebuilt. In addition, entry Fa-5 is stopped at 40', hence the point at 60' in Fig. 2c is incorrect.
Last note on Figure 2: please use colors that are more different from each other! The points are 20 and 25 [Al]/[Nd] are hardly distinguishable!! - Figure 3: where are the peaks corresponding to the other olefinic protons of the side chain of myrcene and farnesene?
- Figure 4a: I have some doubts that the integration areas of the two peaks around 124 ppm are almost equal, bringing to a 43/50 1,4-cis/1,4-trans ratio.
- Paragraph 3.4: more details are required to better understand this section concerning the sequential polymerization. Indeed, it is not clear at which time the second step takes place. Moreover, why did the authors decide to add the same monomer instead of adding a different monomer and obtain a two-block polymer? This would be more instructive and especially useful since CCTP polymerization is especially useful for the synthesis of multiblock copolymers, which are hardly obtainable by other polymerization pathways (see for example Chem Rev 2013, 113, 3836–57)
This paragraph should be expanded, and some other experiments of CCTP of different monomers are required at this point. - Line 402: total yield of 150% is meaningless.
Additional question: what's the point of using CCTP instead of vinyl insertion-coordination polymerization? Which benefits are obtained?